# Untargeted Metabolomic Assay of Prefrail Older Adults after Nutritional Intervention

**DOI:** 10.3390/metabo12050378

**Published:** 2022-04-21

**Authors:** Alina Jaroch, Mariusz Kozakiewicz, Karol Jaroch, Emilia Główczewska-Siedlecka, Barbara Bojko, Kornelia Kędziora-Kornatowska

**Affiliations:** 1Department of Nutrition and Dietetics, Faculty of Health Sciences, Nicolaus Copernicus University in Toruń, Ludwik Rydygier Collegium Medicum in Bydgoszcz, 85-626 Bydgoszcz, Poland; alina.jaroch@cm.umk.pl; 2Department of Geriatrics, Faculty of Health Sciences, Nicolaus Copernicus University in Toruń, Ludwik Rydygier Collegium Medicum in Bydgoszcz, 85-094 Bydgoszcz, Poland; markoz@cm.umk.pl (M.K.); egs@cm.umk.pl (E.G.-S.); kornelia.kornatowska@cm.umk.pl (K.K.-K.); 3Department of Pharmacodynamics and Molecular Pharmacology, Faculty of Pharmacy, Nicolaus Copernicus University in Toruń, Ludwik Rydygier Collegium Medicum in Bydgoszcz, 85-089 Bydgoszcz, Poland; karol.jaroch@cm.umk.pl

**Keywords:** frailty, nutritional status, metabolomics, protein consumption, SPME-LC-HRMS

## Abstract

Frailty is a geriatric syndrome causing a reduction in the body’s functional reserves. Proper nutrition may be helpful in delaying transitioning older adults from pre-frail to frailty syndrome. The present study evaluates the nutritional status of pre-frail patients who underwent nutritional intervention and metabolomic changes resulting from this intervention. Sixteen pre-frail patients (68.4 ± 5.5 years old; 81.3% women) were enrolled for nutritional intervention, and twenty-nine robust elderly people (69.3 ± 5.3 years old; 82.8% women) were the control group. Pre-frail patients consumed 1.0 g protein/kg BW/day for eight weeks through diet modification and an additional daily intake of a protein powder formula. Taken measurements included: Nutritional anthropometry, assessment of food intake, and blood serum analysis with an untargeted metabolomic assessment. Protein consumption increased by 25.8%; moreover, significant increases in body weight (+1.2 kg; *p* = 0.023) and muscle mass index (+0.1 kg/m^2^; *p* = 0.042) were also observed. The untargeted metabolomic assay showed a significant increase in arachidonic acid (*p* = 0.038), and valine (*p* = 0.008) among pre-frail patients. Increased protein consumption is reflected in improved anthropometric and biochemical parameters of pre-frail patients. Moreover, metabolomic assay can be a useful tool in determining compliance with dietary recommendations.

## 1. Introduction

Frailty is a geriatric syndrome first described by Fried et al. It causes a reduction in the body’s functional reserves and, most importantly, the body’s resistance to stressors. Qualification to the frailty syndrome is based on five criteria—shrinking, grip strength, exhaustion, walk time, and physical activity. A coexistence of at least three out of five variables classifies a person as frail, one or two as pre-frail and none as robust [1]. According to expert consensus, frailty is a clinical syndrome that can be reversible or less severe through appropriate nutritional and medical interventions. Particular attention has been paid to the impact of nutritional interventions aimed at the proper calories and protein intake in the diet [2]. Frailty is strongly connected with protein-energy malnutrition, loss of muscle mass, impaired muscle function, and disability [3]. Moreover, with age body composition changes towards higher fat tissue content, and along with insufficient food consumption (especially high quality protein), insufficient synthesis and excessive loss of protein, these factors can lead to the development of malnutrition and sarcopenia. Good nutritional status and sufficient dietary protein intake are required to slow down age-related changes in body composition and maintain good quality of life [4].

Metabolomics is a current and strongly developed “-omics” approach. Its goal is an extensive sample analysis of the low molecular weight molecules from various sample types. Identification of metabolites present in the sample is possible by using advanced analytical approaches, like liquid chromatography−mass spectrometry (LC-MS), followed by computerized processing of the received data [5]. LC-MS, while it is not the only method used in metabolomics, is characterized by a high sensitivity and wider metabolite detection compared to nuclear magnetic resonance or gas chromatography−mass spectrometry [6]. In metabolomics it is possible to perform untargeted and targeted metabolite identification. Typically, data acquired from untargeted analysis is subsequently studied using targeted approaches [5]. Untargeted analysis generates known and unknown metabolites and can be helpful in detailed characterization of biological pathways and in determining novel metabolites—potential biomarkers for diagnosis of present conditions or conditions that are not yet clinically manifested [7]. Metabolomic data acquisition starts with sample preparation and analysis. A beneficial and still improving method of sample preparation is solid phase microextraction (SPME). First described in the 1990s, it is an analytical technique in which a small amount of the extraction phase is immobilized on a thin fiber or other support [8]. With success, it has been applied in many biological matrices [9], enabling to perform an effective untargeted metabolomic assessment. SPME merges the extraction step with metabolism quenching, can determine low molecular weight molecules [10], and has been successfully employed for metabolomic analysis in many studies [11].

We hypothesized that through nutritional intervention focused on increased high-quality protein consumption, pre-frail older adults will improve their anthropometric and biochemical parameters, thereby delaying the development of the frailty syndrome. In order to test this hypothesis the impact of nutritional intervention, conducted among pre-frail patients, was assessed by performing anthropometric measurements and a non-targeted metabolomic analysis of blood serum samples. As a control group, robust patients were also included in the analysis.

## 2. Results

In control and pre-frail groups, the vast majority of study participants were women (>80%) under 75 years old living in cities (Table 1). Pre-frail patients were characterized by weight (*p* = 0.007) and appetite loss (*p* = 0.028), though their BMI values were correct according to ranges used among older adults (BMI = 23.65 kg/m^2^). In the present study, particular attention was paid to protein intake and parameters assessing the state of protein nutritional status. Protein consumption of pre-frail patients was significantly lower (*p* < 0.001) as they did not meet the Recommended Daily Allowances (RDAs) for protein (<0.9 g/kg/bw/day). Among parameters used to assess the protein nutritional status mid-arm muscle circumference (MAMC) and muscle mass index (MMI) differed significantly, while calf circumference (CC) and albumin values were in line with the norm. Moreover, among pre-frail patients weight loss was observed (2.3%).

Twelve pre-frail patients who finished nutritional intervention (four patients withdrew from further participation in the study) were grouped in pairs to evaluate changes in values at baseline and at the end of the study (Table 2). Protein consumption increased from 0.69 g/kg/bw to 0.93 g/kg/bw (25.8%), and exceeded values observed among robust patients. Moreover, a significant increase in body weight (+1.2 kg), MAMC (+0.5 mm) and MMI (+0.1 kg/m^2^) values were observed. The only parameter that decreased after nutritional intervention was albumin concentration (−0.1 g/dL); however, it continued to indicate a proper nutritional status (4.42 g/dL).

In the study, to evaluate the effect of nutritional intervention on the protein nutritional status of pre-frail patients, a control group of robust patients was included. Significant differences were found for body weight (*p* = 0.021), MMI (*p* = 0.011), protein consumption (*p* = 0.0002) and BMI (*p* = 0.007). These parameters were selected for further post hoc analysis (Figure 1). Body weight and protein consumption of pre-frail patients who finished nutritional intervention (PF2) did not differ from robust patients (R), while significant differences were found between other groups.

Principal component analysis (PCA) based on found compounds showed changes in metabolome of examined pre-frail patients who finished the nutritional intervention (*n* = 9, pre-frail patients at baseline—PF-1, black dots; pre-frail patients at the end of nutritional intervention—PF-2, grey dots) (Figure 2). Each dot represents particular patient with unique coordinates that are assigned based on the presence of low molecular weight compounds found within it. The closer dots are to each other, the higher similarity between samples.

After untargeted metabolomic analysis of blood serum samples from nine patients (patients who had residual blood material for further analysis), thirty-one compounds were selected and subjected to further statistical analysis. The selected thirty-one compounds were chosen because of the possible participation in the body’s protein balance or were health markers (Appendix A). Mean values of three compounds were significantly different at the end of the study compared to baseline—arachidonic acid (*p* = 0.038), oleoylethanolamide (*p* = 0.011), and valine (*p* = 0.008). Arachidonic acid and valine values increased, while oleoylethanolamide decreased, which was observed by estimating fold change for those compounds (1.46 ± 0.86, 1.72 ± 1.22, 0.66 ± 0.27, respectively).

Moreover, selected compounds were compared in all groups (robust, pre-frail at baseline, pre-frail at the end of the nutritional intervention). Table 3 shows obtained peak areas for these compounds, which are proportional to amount of the compound that is present in the sample. Significant differences were observed for arachidonic acid, and oleoylethanolamide, but not for valine. Attention should also be drawn to changes in taurine, methionine, and leucine concentration, which after the nutritional intervention were close to or exceeded concentrations obtained by robust patients.

## 3. Discussion

The present study assessed the nutritional status of pre-frail patients who underwent nutritional intervention. The essence of this study is emphasized by the fact that the frailty syndrome can progress rapidly—within 1.5 years, as much as 25% of pre-frail patients can become frail [12]. The incidence of frailty and pre-frail state increases with age [13], and most patients are women. Moreover, low education level, often measured as years of education, is also associated with the frailty frequency [14,15]—all characteristics were observed in our study. In terms of nutritional status, pre-frail patients were at the edge of malnutrition (MNA = 23.6; 2.3% weight loss); still, their BMI (23.65 kg/m^2^) was within the normal range according to the norm developed for older adults (22–27 kg/m^2^) [16]. Weight loss observed among pre-frail patients, regardless of whether it was intended or not, can accelerate muscle loss naturally resulting from the aging process, and further decrease functional capacity for independent living [17,18]. Additionally, nearly all pre-frail patients declared appetite loss, which can lead to protein deficits, and further adverse consequences, including impairment of muscle and skeletal and immune system functions [19]. Impairment of muscle mass, especially development of sarcopenia, will always have a negative effect on the health status of the older adults [20]. Loss of muscle mass is associated with various types of disability, cognitive impairment, as well as with increased mortality, while higher skeletal muscle values and calculated muscle tissue indicators, are associated with the process of normal aging [21].

Pre-frail patients underwent an eight-week nutritional intervention focused on increased protein consumption. Body weight significantly increased (+1.2 kg, *p* = 0.023)—nutritional intervention nearly reversed weight loss observed at baseline (~1.5 kg weight loss). We also observed a significant increase in MAMC and MMI, which may indicate a very favorable increase in muscle mass, possibly preventing the development of sarcopenia [22]. Sarcopenia further contributes to the development of frailty [23], and among older adults promotes malnutrition, and low physical activity [24]. Moreover, it can be concluded that patients followed the proposed diet modification and consumed protein formula—protein consumption increased by 25.8%, and was higher than the RDA (0.93 g/kg BW/day). The importance of implementing nutritional intervention to prevent pre-frailty and frailty progression in older adults has been previously emphasized [25,26], but usually interventions included higher protein doses—about 25–30 g of protein daily [27,28]. Present study showed that even smaller enhancement of daily diet with high quality protein may be beneficial for older people. Currently, a major discussion topic is an indication of a meal to which increased protein supply should be added [19,29].

Metabolomic assessment showed that nutritional intervention caused a significant change in three compounds—arachidonic acid (ARA), oleoylethanolamide (OEA), and valine. ARA is a polyunsaturated *n*-6 fatty acid (20:4) present in phospholipids of body cell membranes; it can be synthesized de novo from linoleic acid [30]. It has a significant role in various metabolic pathways, e.g., it is a precursor for the synthesis of eicosanoids (prostaglandins and leukotrienes) [31]. ARA metabolites, prostaglandins and leukotrienes, that are involved in myogenesis, inflammation, muscle pain, and in chronic inflammatory diseases may play an important role in pathogenesis of muscle weakness and atrophy [32]. Emerging is the role of ARA in skeletal muscle, a site of its retention, where it usually accounts for around 10–20% of phospholipid fatty acids content, and can promote growth and repair of skeletal muscle tissue, especially during resistance training [33]. It was demonstrated in another study that supplementation with 1.5 g of ARA/day in combination with a resistance-training program can increase lean body mass, upper-body strength, and peak power [34]. Moreover, ARA supplementation causes an increase in muscle fatty acids, and plasma lipid ARA content [35]. Observed increase in ARA concentration may be explained by the fact that serum lipid composition is highly sensitive to dietary ARA intake. Food sources of arachidonic acid are simultaneously sources of complete protein—meat, poultry, fish, and eggs [36]—products that we recommended for daily consumption. Further research could indicate if diet modification in combination with resistance training and ARA supplementation would positively affect lean body mass and strength of pre-frail patients. We also observed a significant increase for valine (fold change 1.72, *p* = 0.008). Valine, together with leucine and isoleucine, is a branched-chain amino acid (BCAA), which stimulates muscle protein synthesis after resistance training. This feature was demonstrated in studies on cell and animal models in which enhanced anabolic intracellular signaling was reported in response to isolated BCAA intake [37]. Currently, adequate consumption of high quality protein from animal products, containing all the essential amino acids in proper amounts, seems more important than isolated BCAA supplementation [38]. In the present study, pre-frail elderly were supplied with a protein formula, which was isolated milk protein, and furthermore seniors received individual menus in which animal products were the main source of dietary protein. Presented nutritional intervention may have resulted in increased valine, which could be a potential source for muscle synthesis.

Oleoylethanolamide (OEA) is a bioactive lipid molecule (omega-9 monounsaturated fatty acid; 18:1) derived from oleic acid [39]. OEA stimulates fat catabolism and take part in the control of food intake—it has an anorectic signaling after a high-fat meal [40]. In the present study, OEA decreased after nutritional intervention (fold change 0.56 ± 0.34; *p* = 0.011), while oleic acid increased (fold change 1.18 ± 0.58). It is well known, that circulating *N*-acylethanolamines and their corresponding fatty acids (like OEA and oleic acid) show a positive correlation, and diets high in oleic acid increase circulating OEA [41]. In the present study it would be expected that with the observed increase in oleic acid, values of oleoylethanolamide would also increase. Decrease of OEA can be partially explained by the fact that OEA levels in blood are usually lower than in organs, and its presence in plasma probably results from its outflow of peripheral tissues. Moreover, OEA levels in serum are higher before food consumption and lower after meals [42], and examined patients had their blood collected at a convenient time of the day, regardless of the time of eating.

A study of Pujos-Guillot (2018) focused on the identification of pre-frailty sub-phenotypes using metabolomics; 212 pre-frail and robust patients were selected for untargeted serum metabolomics at baseline, and after a one-year follow-up. Volcano plot analysis of significant ions was made separately for women and men [43]. We analyzed which compounds were also observed in pre-frail patients in the present study. Two essential amino acids, phenylalanine and threonine, were detected in both studies. Like valine, the main food sources of these amino acids are milk and dairy products, meat, fish, poultry, and eggs. Presence of these amino acids in the patient’s serum is associated with the diet consumed, thus adherence to the obtained nutritional recommendations. Low plasma levels of essential amino acids were found in severely frail older people [44]; moreover, reduced concentrations of leucine and isoleucine were observed in older adults with sarcopenia [45]. High concentrations of these amino acids in the present study may have a protective role in the development of malnutrition, sarcopenia, and frailty.

The present study has some limitations. Due to a small number of participants, the study shows preliminary findings, which are to be used in the development of a larger research project. Only telephone contact was provided for pre-frail patients during the nutritional intervention, which may have resulted in patients’ withdrawal. Moreover, the use of only one 24-h dietary recall may be considered insufficient for the assessment of dietary habits. However, it has been shown that even one-day assessment is an appropriate alternative to three-day interviews [46].

## 4. Materials and Methods

This observational prospective study was conducted from March 2015 till August 2017, according to the guidelines laid down in the Declaration of Helsinki. The study was approved by the local Bioethics Committee of the Collegium Medicum, Nicolaus Copernicus University in Torun, Poland (consent No 54/2015). Written informed consent was obtained from all patients. The condition of recruitment was a positive diagnosis of frailty syndrome based on the criteria of Fried et al. and age over 60 years [1]. In the premise of the study, it was necessary to carry out an eight-week nutritional intervention that was eventually not offered to frail patients due to numerous comorbidities, often a severe clinical condition and limited mobility. Therefore, it was decided to recruit pre-frail patients who were eligible to participate in the nutritional intervention program. At baseline anthropometric measurements were taken, also a venous blood sample, and a 24-h food intake recall was conducted to determine the daily nutrient intake, paying particular attention to total protein intake. Blood collection was made at a convenient time of the day, patients were not asked to be in the fasting state. Sixteen patients were qualified to participate in the nutritional intervention and twelve finished the intervention (withdrawal from the study due to lack of time in the result of parental care *n* = 2, no contact with the patient *n* = 2). In order to evaluate compliance with dietary recommendations, the same measurements were made at the end of the study; in particular, 24-h recall was used to control daily protein consumption. In addition, a control group of twenty-nine patients ≥60 years old who did not meet any of the frailty syndrome criteria (robust patients) was formed.

### 4.1. Anthropometric Measurements

Weight was measured using an electronic scale, and patients self-administered height value. Based on these two measurements, the Body Mass Index (BMI) was calculated according to the World Health Organization (WHO) formula. Weight loss during the past three months was calculated using actual and typical body weight. Handgrip strength was measured using the MAP KERN–MAP 80K1S digital dynamometer. The result was the average of three measurements, accurate to one decimal place, expressed in kilograms. Arm (AC) and calf circumference (CC) were measured on the non-dominant side of the body, using anthropometric tape with a millimeter scale SECA 201, giving the result with an accuracy of 0.1 cm. Using a Holtain Skinfold Caliper thickness of the triceps, skinfold was measured with an accuracy of 0.1 mm. Together with AC it was used to calculate mid-arm muscle circumference (MAMC).

To obtain muscle mass index (MMI), first the skeletal muscle mass was calculated using the formula proposed by Lee, which includes body weight, height, gender, age, and race [47]. Next, skeletal muscle mass was divided by height (in m^2^) giving the MMI value [48].

### 4.2. Nutritional Intervention

Pre-frail patients with improper protein intake (<0.9 g/kg BW/day) were offered a dietary intervention. An additional exclusion criterion was included: Lack of diseases limiting protein intake, e.g., severe kidney failure. The intervention consisted of an eight-week modification of food intake: Patients received weekly menus tailored to their tastes and eating habits, and free protein powder formula to enhance their daily diet in protein. The proposed meal plans were designed to ensure the intake of dietary protein at 0.9 g/kg BW/day and the additional protein formula was to be consumed with breakfast and increase protein intake to 1.0 g/kg BW/day. The used protein formula was a medical product (Resource Instant Protein^®^ from Nestlé Health Science, Warsaw, Poland). It was given to study participants for free and patients received the amount of formula needed for eight weeks of intervention at baseline. According to calculations depending on the patient’s body weight, to meet the assumed consumption of 1.0 g protein kg/body weight, patients should have received a minimum of 6 g of protein from the given formula. In order to unify the amount of the formula consumed by all patients and taking into account that protein intake was frequently reduced below recommendations the decision was made to daily supply pre-frail patients with two tablespoons of the protein formula. This provided 10 g of powder per day, which contained 9 g of wholesome animal protein (product made from milk protein, nutritional value: 371 kcal/100 g, 97% kcal from protein). Moreover, all study participants had assured a constant telephone contact with the principal investigator.

### 4.3. SPME Protocol

Blood serum samples were collected at the beginning of the study and at the end of an 8-week follow-up (±3 days). Samples were kept frozen at −80 °C until analysis. All samples were thawed simultaneously and the SPME assay was performed. Mix-mode fibers (7 mm, Supelco, Belafonte, USA) were preconditioned with methanol:water (1:1 *v*/*v*, overnight, static) prior to use. Fibers were washed with water (5 s, static), then directly immersed into 250 μL of serum sample and left for 2 h for extraction of small molecules at 850 RPM (BenchMixer™ XL Multi-Tube, Benchmark Scientific, Sayreville, NJ, USA). After extraction, fibers were quickly rinsed with water (5 s, static) to remove loosely attached matrix components. Compounds adsorbed onto the coating were desorbed with 100 μL of acetonitrile:water solution (4:1, *v*/*v*, 2 h using vortex agitation of 1200 RPM, BenchMixer™ XL Multi-Tube, Benchmark Scientific), with subsequent injection of extracts onto the LC-MS platform for instrumental analysis. Pooled QC and solvent blanks were employed for untargeted metabolomic analysis. All chemicals were MS grade and purchased from Sigma-Aldrich (Poznań, Poland).

### 4.4. LC-MS Analysis

In order to obtain more comprehensive data and cover a wide range of compounds, two separate chromatographic methods were used. Chromatographic separations were performed with reversed phase and HILIC columns: pentafluorophenyl (PFP) Discovery HS F5, 100 mm × 2.1 mm, 3 μm, (Supelco, Bellefonte, PA, USA); mobile phase: A: 99.9% water +0.1% formic acid B: 99.9% acetonitrile +0.1% formic acid, and Luna HILIC, 100 mm × 2.0 mm, 3 μm, 200 A, (Phenomenex, Torrance, CA, USA); mobile phase: A: acetonitrile/ammonium acetate buffer (9:1 *v*/*v*, 20 mM effective salt concentration) B: acetonitrile/ammonium acetate buffer (1:1 *v*/*v*, 20 mM effective salt concentration). The LC gradient for the PFP column was 0–3 min 0% B, 3–25 min linear gradient to 90% B, 25–34 min 90% B, 34–40 min 0% B, and a flow of 0.3 mL × min^−1^. The HILIC column gradient was set as follows: 0–3 min 0% B, 3–8 min linear gradient to 100% B, 8–12 min 100% B, 12–20 min 0% B, and a flow of 0.4 mL × min−1. Both gradients were adopted from a previous work of Vucovic and Pawliszyn [49].

The LC-MS system consisted of a Dionex UltiMate 3000 RS autosampler, Dionex Ultimate 3000 RS pump (Thermo Fisher Scientific, Bremen, Germany), and a Q-Exactive Focus high resolution mass spectrometer (Thermo Fisher Scientific, Bremen, Germany). The MS was operated in both positive and negative ionization modes for each chromatography mode. The reverse chromatography in positive ionization mode was run with HESI ion source parameters set as follows: Spray voltage 1500 V, capillary temperature 300 °C, sheath gas 40 a.u., aux gas flow rate 15 a.u., probe heater temperature 300 °C, S-Lens RF level 55%. For reverse chromatography in negative ionization mode, HESI ion source parameters were as follows: Spray voltage 1500 V, capillary temperature 256.25 °C, sheath gas 47.5 a.u., aux gas flow rate 11.25 a.u., probe heater temperature 412.5 °C, S-Lens RF level 55%.

HILIC chromatography in positive ionization mode was run with HESI ion source parameters set as follows: Spray voltage 1500 V, capillary temperature 320 °C, sheath gas 60 a.u., aux gas flow rate 30 a.u., aux gas flow rate 21 a.u., probe heater temperature 300 °C, S-Lens RF level 55%. For HILIC chromatography in negative ionization mode, HESI ion source parameters were as follows: Spray voltage 1500 V, capillary temperature 325 °C, sheath gas 60 a.u., aux gas flow rate 30 a.u., probe heater temperature 425 °C, S-Lens RF level 55%. Data acquisition was performed by Xcalibur software v. 4.0 (Thermo Fisher Scientific, Bremen, Germany).

### 4.5. Metabolomics Data Processing

Compound Discoverer 2.1 (Thermo Fisher Scientific, Bremen, Germany) was utilized for metabolomics data processing. The selected mass tolerance window was set up to 3 ppm, signal-to-noise threshold to 1.5, max sample-to-blank ratio >5, retention time max shift 2 min, and min peak intensity 10,000. Data were subjected to auto-scaling and the pooled QC-based area was used for correction (min 50% coverage, max 30% RSD in QC, normalization by constant mean). Features of characteristic *m*/*z* found in samples were annotated, describing their compounds, and their biological meaning was searched for in the Human Metabolome Database (HMDB). The initial *m*/*z* values of preselected compounds were back-searched once more in the unprocessed raw files of metabolomics data.

### 4.6. Statistical Analysis

Statistical analyses were performed at the 0.05 significance level (*p* < 0.05). Normal distribution was tested using the Shapiro–Wilk test. In the assessment of differences in the average level of the numerical variables in two populations, the Mann–Whitney test was used, while in the analysis of variables having the character of qualitative data, the Pearson’s chi-squared test was used. Student’s t-test for dependent variables or Wilcoxon signed-rank test was used to assess the size of the difference between pre-frail patient pairs (end of the study/baseline). For more than two populations (robust, pre-frail baseline, pre-frail end) one-way ANOVA was used, and its non-parametric equivalent—Kruskal–Wallis *H* test. Furthermore, post hoc analysis was made to assess precisely between which groups significant difference was observed using Tukey’s test.

Statistical analysis was performed using STATISTICA 12.5 (version 12.5, StatSoft, Palo Alto, CA, USA).

## 5. Conclusions

In the present study, pre-frail older adults underwent a nutritional intervention focused on increased protein intake. The nutritional status of pre-frail patients improved, as they increased their protein consumption by over a quarter. Thus, their values of anthropometric parameters and protein consumption were similar to robust patients, making the intervention potentially helpful in reducing the risk of developing sarcopenia and frailty. Untargeted metabolomic analysis of blood serum samples showed significant increase in valine and arachidonic acid concentration, suggesting compliance with dietary recommendations, and moreover implying that metabolomic assay can be a useful tool to determine compliance during nutritional intervention studies.

## Figures and Tables

**Figure 1 metabolites-12-00378-f001:**
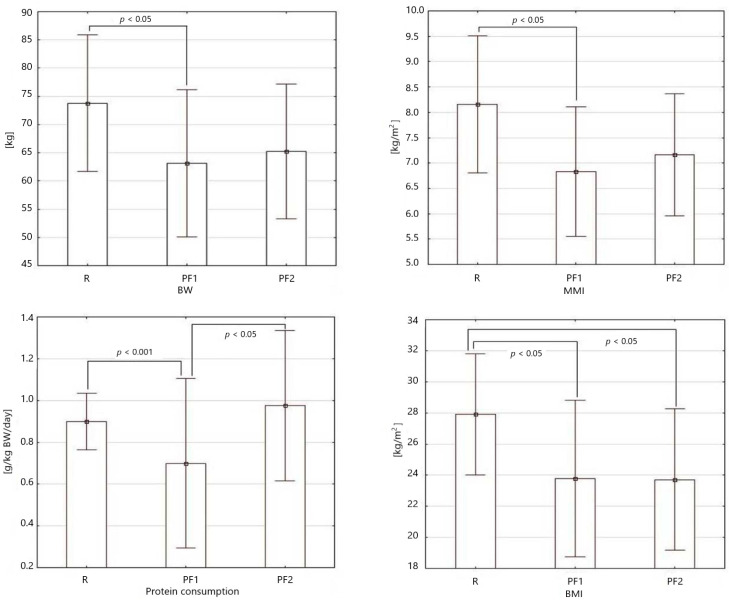
Bar graphs of body weight (BW), muscle mass index (MMI), protein consumption, and BMI among robust (R), pre-frail at baseline (PF1), and pre-frail patients at the end of the study (PF2), showing statistical differences according to post hoc analysis.

**Figure 2 metabolites-12-00378-f002:**
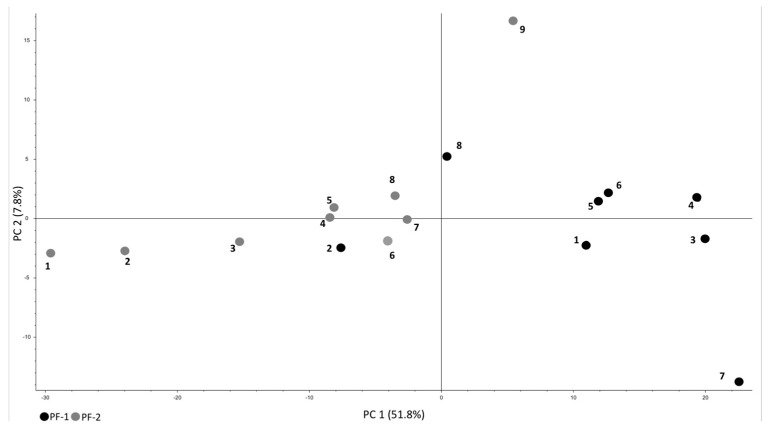
Principal component analysis (PCA) of observed metabolome changes among pre-frail patients at baseline (PF-1, black dots) and at the end of nutritional intervention (PF-2, grey dots).

**Table 1 metabolites-12-00378-t001:** Basic characteristics of study participants.

	Control *n* = 29	Pre-Frail *n* = 16	*p*
Gender: women	82.8%	81.3%	0.899
Age, y	69.3 ± 5.3	68.4 ± 5.5	0.618
Age			0.618
60–75	87.5%	82.8%	
>75	12.5%	17.2%	
Residence			0.934
village	6.9%	6.3%	
city	93.1%	93.7%	
Education			0.642
basic	17.3%	25.0%	
secondary	51.7%	37.5%	
higher	31.0%	37.5%	
Appetite loss: yes	25.0%	96.5%	0.028
Body weight, kg	73.8 ± 12.1	65.0 ± 14.0	0.034
Weight loss, %	-	2.3 ± 5.0	0.007
BMI, kg/m^2^	27.91 ± 3.88	23.65 ± 4.95	0.009
Handgrip, kg (w)	23.0 ± 5.3	22.1 ± 4.9	0.593
CC, cm	36.3 ± 3.0	34.6 ± 3.0	0.086
MAMC, mm (w)	23.0 ± 2.8	22.3 ± 3.4	0.048
MMI, kg/m^2^ (w)	7.7 ± 1.4	6.8 ± 1.5	0.048
Albumin, g/dL	4.64 ± 0.22	4.50 ± 0.46	0.571
MNA	27.0 ± 1.2	23.6 ± 3.4	<0.0001
Protein consumption, g/kg/bw	0.9 ± 0.14	0.71 ± 0.37	<0.001

*p*—statistical significance (<0.05), (w)—results for women, CC—calf circumference, MAMC—mid-arm muscle circumference, MMI—muscle mass index, MNA—mini-nutritional assessment.

**Table 2 metabolites-12-00378-t002:** Observed change in anthropometric and protein balance parameters among pre-frail patients who finished nutritional intervention (*n* = 12).

	Difference	*p*
Body weight, kg	+1.2 ± 1.6	0.023
BMI, kg/m^2^	+0.4 ± 0.6	0.027
Handgrip, kg	+0.6 ± 1.9	0.272
CC, cm	+0.2 ± 0.6	0.269
MAMC, mm	+0.5 ± 0.7	0.028
MMI, kg/m^2^	+0.1 ± 0.2	0.042
Albumin, g/dL	−0.1 ± 0.3	0.169
Protein consumption, g/kg/bw	+0.24 ± 0.17	0.002

*p*—statistical significance (<0.05), CC—calf circumference, MAMC—mid-arm muscle circumference, MMI—muscle mass index.

**Table 3 metabolites-12-00378-t003:** Observed peak area of selected compounds among robust and pre-frail patients (at baseline, and at the end of study).

Compound	Robust (×10^6^)	Pre-Frail Baseline (×10^6^)	Pre-Frail End (×10^6^)	*p*
Arachidonic acid	3.5 ± 1.3	2.8 ± 0.7	4.1 ± 1.3	0.037
Oleoylethanolamide	25.2 ± 6.3	27.6 ± 7.3	15.5 ± 9.9	0.002
Valine	0.41 ± 0.42	0.35 ± 0.26	0.62 ± 0.38	0.147
Taurine	4.2 ± 2.7	3.5 ± 2.0	5.0 ± 2.3	0.068
Methionine	11.1 ± 3.0	9.0 ± 3.1	11.5 ± 2.2	0.075
Leucine	190.7 ± 27.0	168.8 ± 24.0	187.9 ± 22.0	0.066

*p*—statistical significance (<0.05).

## Data Availability

Not applicable.

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
