# Peer review of "Untargeted Metabolomic Assay of Prefrail Older Adults after Nutritional Intervention"

_metabolites, 2022, doi:10.3390/metabo12050378_

Round 1

Reviewer 1 Report

The authors used metabolomics to evaluate the effect of nutritional intervention on prefrail older adults. 16 pre-frail patients were given a higher amount of protein in their diet, while 29 robust elderly adults were used as control without diet intervention. After 8 weeks of diet intervention, changes such as increased body weight, BMI, MAMC and MMI were observed. However, albumin concentration was found to decrease. Several metabolites, ie, arachidonic acid, oleoylethanolamide, valine, taurine, methionine and leucine were found to increase after the diet intervention.

The study is interesting but suffers from inappropriate control and analysis protocol. The true control in this study should be prefrail elderly adults, not the robust elderly. They should be randomly selected and should not undergo diet intervention. If the authors wished to compare the same group of prefrail elderly before and after diet intervention, then the robust elderly control has no meaning.

Another concern I have is the sample preparation. Why was SPME used in this study? Binding of small molecules to the SPME fiber is compound and concentration dependent. This can result in a bias in favor of metabolites that have higher affinity to the fiber or are most abundant. The small number of metabolites identified (listed in Table S1) shows that SPME protocol is not appropriate. Obviously, it is not an untargeted metabolomics assay which the authors claim in the manuscript title.  

Resolution of Figures 1 and 2 need to increase. Figure 1 legend is missing MMI. In addition, Figure 1 appears to contract Table 2 if I understand the legend of Figure 1 correctly. Figure 1 shows no significant difference between PF1 (which I think is the PF group at the beginning of intervention) and PF2 (which I think is the PF at the end of intervention). However, both BMI and MMI show significant changes in Table 2.

Reviewer 2 Report

This manuscript describes an interesting study on nutritional implications of prefrail older adults by anthropometric and biochemical evaluations. The manuscript is well organized, an adequate analytical and methodological approach has been carried out, and the data has been correctly presented and discussed by the Authors. This investigation provided new knowledge and important features for future research in this area.

Overall the manuscript is easy to understand, however some points must be corrected.

Pg 8 lines 263-265: the Authors reported that thirty-nine compounds were selected and subjected to statistical analysis. What are these compounds? Only 6 compounds are listed in Table 3! Please specify.

Pg 8 lines 267-269: the Authors reported that the mean values of four compounds were significantly different……..but only three compound were mentioned, that are arachidonic acid, oleoylethanolamide and valine. Please correct.

Although the paper has some limitations as noted by the Authors, there are no particular issues to point out, and in my opinion the manuscript can be accepted for publication without any substantial changes as suggested.

Round 2

Reviewer 1 Report

The authors have addressed my concerns. In my opinion, the manuscript can be accepted for publication after the font size for X- and Y-Axis on Figure 2 are increased.